# The Nuremberg Address Knowledge Graph

Oleksandra Bruns[1,2], Tabea Tietz[1,2], Mehdi Ben Chaabane[2], Manuel Portz[2], Felix Xiong[2], and Harald Sack[1,2]

[1] FIZ Karlsruhe – Leibniz Institute for Information Infrastructure, Germany
`firstname.lastname@fiz-karlsruhe.de`
[2] Karlsruhe Institute of Technology, Institute AIFB, Germany

**Abstract.** The research of European history across various time layers gives insights about the development of the European cultural identity. Nuremberg as one of the great European metropolises during the Middle Ages experienced a number of transformations throughout the centuries. Within the TRANSRAZ research project, Nuremberg and the development of its architecture and culture is recreated from the 17th to the 21st century. It will be available for researchers and the public by means of an interactive 3D environment. Goal of this poster paper is to discuss the ongoing work of connecting heterogeneous historical data from sources previously hidden in archives to the 3D model using knowledge graphs for a scientifically accurate exploration of Nuremberg. The contribution of this paper is the Nuremberg Address Knowledge Graph (NA-KG) which contains information of people and organizations in Nuremberg from unstructured data of Nuremberg address books.

**Keywords:** Knowledge Graphs · Cultural Heritage · History.

## 1 Introduction

The exploration of European cities and their development throughout history contributes to our understanding of the European cultural identity. However, to explore our city histories including the cultures within on the Web, historical records have to be collected and transformed into a structured format which can be integrated into intelligent user interfaces. Nuremberg was one of the great European metropolises in the Middle Ages and beyond. Since then, the city experienced numerous transformations, including the almost complete destruction during the Second World War. Therefore, a systematic and scientific reconstruction of the city in different time periods is necessary to preserve this important part of the European cultural heritage and make it accessible for research. Nuremberg's reconstruction was first initiated with the TOPORAZ project [2], in which a virtual research environment (VRE) was created linking a scholarly sound 3D model of the main market of the city of Nuremberg to a database in four different time layers. In this paper, parts of the ongoing project TRANSRAZ [3] (a successor of TOPORAZ) are presented. As part of TRANSRAZ, the Nuremberg History Knowledge Graph is being developed and

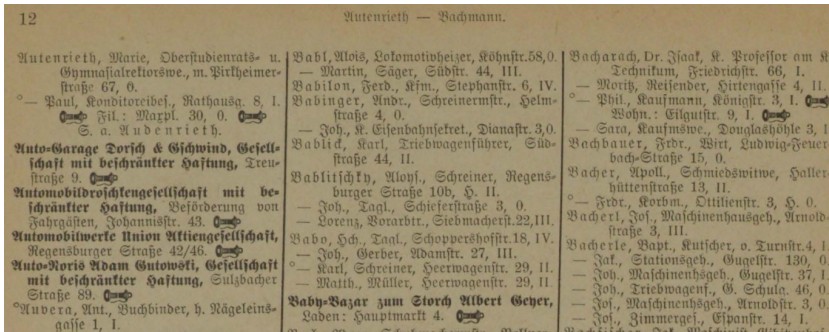

Fig. 1: A digitized page from "Addressbuch von Nürnberg 1910"
.

integrated into the 3D VRE for exploration by researchers and the public. In order to provide means of exploration for Nuremberg, historical resources have to be collected and integrated into the KG. Address books provide one of the most comprehensive and valuable sources to extract knowledge about Nuremberg's citizens across different time periods. The Nuremberg address books date back to 1792 and are available annually since 1890. They form one part of a number of resources used to create the *Nuremberg History Knowledge Graph.* Prior projects integrating historical data into user interfaces to explore city histories include Amsterdam Time Machine[3] and the Time Traveler Berlin application [4]. Furthermore, notable recent efforts involving the creation of Linked Data resources from unstructured OCR data in the cultural heritage domain include [1] and [5].

Goal of this poster paper is to report on the ongoing work of integrating heterogeneous historical data sources into the *Nuremberg History Knowledge Graph* as part of the TRANSRAZ research project. The contribution of this paper is the *Nuremberg Address Knowledge Graph (NA-KG)* containing 860K triples on persons and organizations in the Nuremberg address book of 1910. NA-KG is publicly available[4]. This contribution is valuable for historical scientists and digital humanists intending to study citizens' names, historical occupations as well as companies and their distribution in Nuremberg. After its integration into the 3D VRE, the address book data can be explored by researchers as well as the general public.

## 2   Nuremberg Address Knowledge Graph

This section presents the main contribution of the paper and discusses the workflow to populate the *Nuremberg Address Knowledge Graph (NA-KG)* as a part of the *Nuremberg History Knowledge Graph.*
**Data.** The NA-KG is based on data from the 1910 address book of Nuremberg. The book contains information about 1) historical residents, e.g. their names, occupations, addresses, granted civil rights, 2) historical organizations, e.g. names

---

[3] https://amsterdamtimemachine.nl/
[4] https://github.com/ISE-FIZKarlsruhe/Transraz

and types of companies, ownership information and addresses. Starting point are scanned images of the address book in JPEG format (see Figure 1) and their transcribed version in plain TXT format. Due to low paper quality, distortion of pages, poor inking, historic gothic fonts, ligatures, archaic terms and typos in the original sources, the resulting transcribed documents are rather noisy and require significant error correction as described below.

**Segmentation.** Based on the syntactic structure of the text, the data was divided into person entries and company entries. By developing a set of regular expressions the entries were further segmented into their individual components, e.g. last name, company name, street name, occupation. The missing information, e.g. omitted last names in case of namesakes and family members, was inserted.

**Normalization.** The errors in the individual components of the entries were resolved by leveraging reference vocabularies and lookup matching of potential candidates in the lexicon for correction, based on the Levenshtein distance. For addressing orthographic errors in last and first names a list of all first names of Cologne[5] and indexed data from Books of Nuremberg's Twelve Brothers[6] were exploited. For correcting the occupations the lists of German occupations[7] and German historical occupations[8][9] were extracted. The street names were normalized according to the urban space that is covered by TRANSRAZ project.

**The Nuremberg Address Knowledge Graph.** The NA-KG builds upon OWL[10], VCARD[11], FOAF[12], SCHEMA[13] and DBPEDIA[14]. It contains 5 classes: *vcard:Individual, vcard:Address, vcard:Organization, schema:Occupation* and *dbo:Street*; 6 object properties that describe relations between instances of these classes, e.g. *dbo:owningOrganization* and *vcard:hasAddress*; and 10 datatype properties that provide information about instances, e.g. *vcard:organization-name* and *transraz:abbreviatedName*. The NA-KG includes 860K RDF triples based on 165K entities that were extracted from the address book. It consists of structured data about 1403 historical companies and 67K residents of Nuremberg.

Being based on one of the fullest sources of Nuremberg citizens in 1910, the NA-KG makes it possible to develop hypotheses and draw reliable conclusions of the population and subsequently to extend and to enrich the knowledge about life in historical Nuremberg. For example, the existing KG is able to answer the following questions: *What was the ratio of persons with civil rights to persons without civil rights?* (Figure 2), *What was the most common profession in 1910?* (Figure 3),

---

[5] https://offenedaten-koeln.de/dataset/vornamen

[6] https://hausbuecher.nuernberg.de/index.php?do=page&mo=5

[7] https://de.wikipedia.org/wiki/Kategorie:Beruf

[8] https://de.wikipedia.org/wiki/Kategorie:Historischer_Beruf

[9] https://www.guenteroppitz.at/berufe/alte-berufsbezeichnungen-in-matzleinsdorf/

[10] https://www.w3.org/OWL/

[11] https://www.w3.org/TR/vcard-rdf/

[12] http://xmlns.com/foaf/spec/

[13] https://schema.org/

[14] https://www.dbpedia.org/

```
PREFIX transraz: <http://transraz/addressbook#>
PREFIX rdf: <http://www.w3.org/1999/02/22-rdf-syntax-ns#>
PREFIX vcard: <http://www.w3.org/2006/vcard/ns#>
SELECT ?civilRight (COUNT(?civilRights) as ?counter_cvR)
WHERE { ?Individual rdf:type vcard:Individual   ;
            transraz:civilRights ?civilRights   . }
GROUP BY ?civilRights
ORDER BY DESC (?counter_cvR)
```

Fig. 2: The SPARQL query targeting the number of citizens who were granted civil rights.

```
PREFIX rdfs: <http://www.w3.org/2000/01/rdf-schema#>
PREFIX schema: <http://schema.org/>
SELECT ?name (COUNT(?name) as ?frequency)
WHERE { ?Individual schema:hasOccupation ?Occupation    .
        ?Occupation rdfs:label ?name     .}
GROUP BY ?name
ORDER BY DESC (?frequency)
```

Fig. 3: The SPARQL query targeting the frequency of occupations mentioned in "Addressbuch von Nürnberg 1910".

```
PREFIX rdfs: <http://www.w3.org/2000/01/rdf-schema#>
PREFIX schema: <http://schema.org/>
SELECT ?Streetname (COUNT(?Streetname) as ?frequency)
WHERE {?Organization rdf:type vcard:Organization    ;
                    vcard:hasAddress ?AddressID    .
       ?AddressID vcard:hasStreetAddress ?Street    .
       ?Street rdfs:label ?Streetname   . }
GROUP BY ?Streetname
ORDER BY DESC(?frequency)
```

Fig. 4: The SPARQL query targeting the amount of companies for every street in Nuremberg.

or *What was the central business district of Nuremberg? (Figure 4)*[15]. Thus, the provided address book data allows to explore not only the whereabouts of Nuremberg inhabitants but also their social status, family relations and business developments.

---

[15]See    more    example    SPARQL    queries    under    https://github.com/ISE-FIZKarlsruhe/Transraz/tree/main/NurembergAddressKG/SPARQL_Queries

## 3   Discussion

Although the NA-KG is part of ongoing work to create a scientifically sound Nuremberg History Knowledge Graph, the contribution of the first version of NA-KG already provides a valuable structured resource for researchers in historical science, digital humanities and cultural heritage in general.

However, the current version of NA-KG also has shortcomings which will be addressed in future work. For example, in the address books, names and occupations are abbreviated inconsistently which complicates the normalization of the data. Furthermore, the NA-KG will be populated with data from address books of different time periods, thus, entity disambiguation and entity linking across different time layers will be required. To accomplish this, the historical data from address books will be enriched by mapping the entities to external data sources like Wikidata and national authority files. Moreover, for future exploration of such temporal changes the reasonable representation of time component will be considered.

## 4   Conclusion

This paper presents the *Nuremberg Address Knowledge Graph* which contains persons and organizations in the historical city of Nuremberg based on scanned and transcribed address book data. NA-KG is a substantial part of an ongoing work towards a *Nuremberg History Knowledge Graph* aiming to connect heterogeneous historical data about Nuremberg to a 3D virtual research environment. The presented KG allows to draw conclusions about the citizens and businesses of Nuremberg in 1910 and is available publicly on the Web.

**Acknowledgement.** This work is funded by the Leibniz Association under project number SAW-2020-FIZ KA-4-Transraz.

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
