# OpenReview forum: "The Nuremberg Address Knowledge Graph"
_eswc-conferences.org/ESWC/2021/Conference/Poster_and_Demo_Track — ESWC2021 P&D_

### Official Review · AnonReviewer4 · 2021-04-06
**Very practical work. It is valuable but there is some room to improve.**

**Rating:** 6
**Confidence:** 3

**Review:**

It is a practical work where the information in the historical address book in a city is represented as a knowledge graph. The process to generate it from the resource and the abstract structure is explained in the paper.
As a poster in the semantic web conference, it is desirable to explain why the knowledge graph is chosen to represent the given information. In other words, what is the benefit of the knowledge graph in the case? The authors would show (expected) use cases, e.g.,  how it can be used in other research or in applications.
There are some rooms to improve the work from the technical viewpoint. As the mentioned in Discussion, the value of the knowledge graph is interoperability with other knowledge graphs.  There are two issues to assure it. One is the identifier issue, i.e., it is important to make it clear to give identifiers for persons and companies. If the rule for identifying (disambiguating) instances is clear, it is easier to connect instances in other knowledge graphs. The other is the standardization issue. The authors mentioned the occupations are identified by referring to the resources. If the occupation is organized as an ontology or vocabulary, it would give more interoperability with other knowledge graphs.

**Anonymity:**

Yes, I would like my review to remain anonymous.

---

### Official Review · AnonReviewer1 · 2021-04-14
**Thorough poster on the Nuremberg Address Knowledge Graph with potential linkability value**

**Rating:** 7
**Confidence:** 5

**Review:**

In this poster, the authors discuss ongoing work in using knowledge graphs to connect heterogeneous historical data from previously inaccessible archive sources to 3D models. In particular, they contribute the Nuremberg Address Knowledge Graph (NA-KG), containing information of people and organisations in the Nuremberg of 1910 from the structured data of Nubremberg address books. The paper describes its construction process and some of its features, like the vocabularies it reuses, its classes, and its properties. A plan for addressing known shortcomings, population, linking to external sources, and evaluation is included.

The paper is clear and well written, and provides sufficient detail to assess the contribution of the poster. The resulting KG is interesting and can be a valuable resource, especially when integrated and enriched with existing ecosystems of ontologies and RDF graphs in digital humanities and history (see e.g. https://github.com/CLARIAH/awesome-humanities-ontologies -- disclaimer: I pushed the first commit of this one).

I only have minor concerns about the modelling of occupations, the actual contribution, and privacy of the included persons. Regarding the modelling of historical occupations, this has been specifically covered in HISCO (https://historyofwork.iisg.nl) by integrating historical occupational titles in various projects; I would recommend considering this next to schema:Occupation. Regarding the actual contribution, the authors say that the KG “makes it possible to develop hypotheses and draw reliable conclusions…” but I think the examples that follow were already possible without it being a KG? I thought the interesting contribution here is the possibility to enrich and extend knowledge that is related to the history of the city. Finally, it would be good to clarify if persons from 1910 are not subject to privacy regulations and are safe to publicly disclose.

**Anonymity:**

Yes, I would like my review to remain anonymous.

---

### Official Review · ~David_Chaves-Fraga1 · 2021-04-14
**The value of KGs in historical science and digital humanities**

**Rating:** 7
**Confidence:** 5

**Review:**

This paper presents the construction of a new Knowledge Graph for Nuremberg’s addresses. The paper is well written and easy to follow and describes a relevant contribution useful for digital humanities and related sciences. I have three main comments that, IMO, can improve the quality of the paper:
1) Is the workflow follow to construct the KG reproducible? which (semantic web) technologies are being used? Instead of figure 1, I would expect a figure showing the proposed workflow and what are the tools used to construct them. Additionally, an image of the vocabulary used for the KG could help to understand the instances (uploaded to GitHub repository would be enough)
2) Are the steps proposed based on any state-of-the-art approach? I miss references.
3) Follow good open science practices: give license to the data and documentation uploaded to Github in order to understand if they can be reused. Improve the repository including the used tools and (if it is possible) the txt from the input sources. And provide a link in the paper to the project website, I finally found it going via Github repo.

**Anonymity:**

No, I would like my review to be deanonymized.

---

### Decision · Program_Chairs · 2021-04-19

Accept